# The Molecular Biology of Soft Tissue Sarcomas: Current Knowledge and Future Perspectives

**DOI:** 10.3390/cancers14102548

**Published:** 2022-05-22

**Authors:** Julien Vibert, Sarah Watson

**Affiliations:** 1INSERM U830, Équipe Labellisée Ligue Nationale Contre le Cancer, Diversity and Plasticity of Childhood Tumors Lab, Institut Curie Research Center, PSL Research University, 75005 Paris, France; julien.vibert@curie.fr; 2Department of Medical Oncology, Institut Curie Hospital, 75005 Paris, France

**Keywords:** soft tissue sarcomas, molecular biology, targeted therapy

## Abstract

**Simple Summary:**

Over the past 20 years, significant advances in the field of genetics and molecular biology have led to the dismantlement of multiple subtypes of sarcomas. As a result, molecular approaches nowadays play a critical role in the diagnosis, prognostic classification, and therapeutic management of numerous mesenchymal tumor subtypes. This review of the current literature illustrates the main uses of molecular biology in the field of soft tissue sarcomas and the future challenges that remain to be addressed.

**Abstract:**

Soft tissue sarcomas are malignant tumors of mesenchymal origin, encompassing a large spectrum of entities that were historically classified according to their histological characteristics. Over the last decades, molecular biology has allowed a better characterization of these tumors, leading to the incorporation of multiple molecular features in the latest classification of sarcomas as well as to molecularly-guided therapeutic strategies. This review discusses the main uses of molecular biology in current practice for the diagnosis and treatment of soft tissue sarcomas, in addition to perspectives for this rapidly evolving field of research.

## 1. Introduction

Soft tissue sarcomas comprise all malignant tumors that develop from soft tissues in the body and that are thought to derive from a mesenchymal origin. They are mostly rare tumors and characterized by a large clinical and biological heterogeneity, with more than 100 different subtypes in the latest WHO classification [1]. Their management is therefore complex and historically based on histological characteristics, but it has been transformed by the help of molecular biology for diagnosis and treatment [2]. Indeed, while sarcomas can be classified according to their microscopic appearance and probable mesenchymal cell-of-origin, it has become clear that many subtypes are characterized by specific genomic alterations that not only help to refine the diagnostic process but most importantly to understand their oncogenesis, which is paramount for prognosis and treatment. Accordingly, the latest WHO classification puts ever greater emphasis on molecular alterations in the definition of some subtypes of sarcomas [1]. In this review, we will discuss the current state-of-the-art for molecular diagnosis and treatment of soft tissue sarcomas, with a focus on well-established techniques but also on near future perspectives for this rapidly changing and promising field of molecular biology of sarcomas.

## 2. Molecular Biology for Sarcoma Diagnosis

Cancer diagnosis is classically based on pathology, with the consequence that cancers are usually classified according to their organ and/or supposed tissue of origin. However, cancer is primarily a genetic disease, and it has become clear that pathologically homogeneous cancers can harbor a large heterogeneity in their underlying genetic make-up. Since the genetic alterations leading to oncogenesis are determining for the behavior of the tumor, it has become increasingly essential to characterize them for better diagnosis, let alone prognosis, and potentially treatment guidance.

This is no exception for soft tissue sarcomas: the classification is historically based on histological characteristics, but molecular biology has allowed the refinement of the diagnostic nosology of this large and heterogenous group of tumors. For simplicity, sarcomas are classically divided into two groups based on genomic characteristics: (1) sarcomas with a single driver molecular alteration (or sarcomas with “simple genetics”) and (2) sarcomas with a complex genomic profile (sarcomas with “complex genetics”) [3]. The former group comprises sarcomas that are defined by specific driver molecular alterations, mainly oncogenic gene fusions, but also activating or inactivating mutations, or gene amplifications. Therefore, their overall genomic profile is usually “simple” with near-diploid karyotypes, meaning that there are few other genomic alterations other than the driver alteration. If the oncogenic properties of all the gene fusions found in rare sarcomas have not yet been assessed in relevant models, their similarities in terms of structure, the homogeneity of the gene expression profiles of tumors with a given fusion, as well as the scarcity of other genomic alterations found in their genomes, suggest that these molecular alterations are a very early driver event in the oncogenesis of these tumors. This contrasts with the second group of sarcomas which harbor highly rearranged genomic profiles, with large numbers of chromosomal and copy number alterations as well as point mutations including of tumor-suppressor genes, often reflecting genomic instability. This binary classification is probably oversimplifying, and it may be misleading, for instance dedifferentiated liposarcoma is characterized by a driver alteration (*MDM2* amplification), but it also has a highly rearranged genomic profile [4].

For the group of sarcomas with a driver alteration, molecular biology is logically essential for their accurate diagnosis and characterization. For other sarcomas, it also has the potential to inform diagnosis, especially as a useful tool to distinguish them from morphologically similar benign tumors.

## 3. Sarcomas with “Simple Genetics”

Sarcomas with a simple genetic driver alteration represent 30% to 40% of soft tissue sarcomas. They are characterized by specific molecular alterations that are usually pathology-defining, therefore molecular biology is essential to make the diagnosis. Classically, these molecular alterations are divided into oncogenic gene fusions, activating and inactivating point mutations, and gene amplifications.

### 3.1. Gene Fusions

The most common driver alterations in sarcomas are gene fusions. A large number of sarcomas are translocation related, i.e., the result of a chromosomal translocation giving rise to a fusion gene encoding an oncogenic fusion protein, usually a chimeric transcription factor [5]. The paradigm of this model of oncogenesis is Ewing sarcoma [6]: this tumor which develops from bone but also soft tissues in young adults and adolescents is characterized by a translocation between chromosomes 11 and 22, giving rise to a fusion gene *EWS-FLI1*, leading to a chimeric transcription factor with oncogenic properties [7]. In recent years, dozens of other sarcoma-defining gene fusions have been described, thus extending the number of subtypes of oncogenic fusion-driven sarcomas and refining the classification of often similar-looking but biologically different tumors. Most gene fusions involve transcription factors, though some may lead to constitutive activation of a tyrosine kinase receptor or growth factor (Table 1).

In clinical practice, diagnosis of the oncogenic fusion is done using molecular techniques such as fluorescence in situ hybridization (FISH), reverse transcription–polymerase chain reaction (RT-PCR), or targeted RNA sequencing [8]. The former detects rearrangement of genes involved in the fusion, while RT-PCR and targeted RNA sequencing search for the resulting RNA transcript in tumor cells. While both methods are highly sensitive, specific, and accessible in most routine labs, they are targeted assays, and they require a good *a priori* knowledge of the differential diagnoses.

In contrast, a more recent technique based on next-generation sequencing and increasingly used for diagnosis of sarcomas is whole transcriptome profiling (RNA sequencing, RNA-seq). Using this unsupervised technique, a single assay can detect every possible gene fusion leading to a fusion transcript, including yet undescribed oncogenic fusion transcripts. In addition to its powerful fusion detection capacity, profiling the whole transcriptome enables refining, and it helps in classification using transcriptomic similarity to other sarcomas. In this way, novel entities with homogeneous transcriptomic profiles and specific gene fusions have been described. For instance, Watson et al. used RNA-seq to characterize a group of 180 sarcomas for which no diagnosis could be made using FISH or RT-PCR [9]. A gene fusion was detected in more than half of cases, including several previously uncharacterized fusion transcripts. Moreover, whole-transcriptome profiling allowed high-dimensional clustering of sarcomas, showing that most fusion genes are associated with a characteristic transcriptomic profile, and that some sarcomas with differing fusion transcripts can be grouped into transcriptomically homogeneous entities, such as *CIC*-fused sarcomas which comprise *CIC-DUX4*, *CIC-FOX4*, and *CIC-NUTM1* sarcomas. Thus, transcriptomic profiling, and more generally molecular profiling, allows a grouping of sarcomas that may differ from simple pathological diagnosis or gene fusion detection: one can envision that techniques such as RNA-seq could lead to a novel classification of sarcomas complementary of the present pathologically oriented classification. Indeed, some centers such as the Institut Curie are using RNA-seq to help in the diagnosis of sarcomas, primarily for gene fusion detection but also for transcriptomic clustering. Of note, whereas initial use of RNA-seq was restricted to fresh frozen tissues, it has now evolved and can also be performed on paraffin-preserved tissues [10]. RNA-seq has since allowed the characterization of novel fusion genes such as *CIC-NUTM1* [11], *TFCP2*-rearranged [12], *EWSR1-SSX1* [13], as well as the identification of *NTRK*-rearranged sarcomas [14] or *NRG1*-fused sarcomas [15]. It has also led to the identification of different molecular subgroups of entities previously considered as pathologically homogeneous, for instance pediatric and spindle cell rhabdomyosarcomas [16,17]. These molecular alterations defining homogeneous groups of sarcomas have mostly been integrated in the current classification scheme as an essential complementary information to pathology [1].

### 3.2. Mutations

While gene fusions constitute the most frequent molecular alterations in sarcomas, some subtypes are characterized by mutations of specific genes, either oncogenesis “driver” genes (activating mutations), or tumor suppressor genes (inactivating mutations).

#### 3.2.1. Activating Mutations

Though rare in the number of subtypes, some sarcomas present activating mutations in “driver” genes as their primary oncogenic mechanism. The paradigm of this are gastrointestinal stromal tumors (GISTs) that are characterized by gain-of-function mutations of the *KIT* gene (85%), and less often the *PDGFRA* gene (5%), which are both mutually exclusive and lead to constitutive activation of these transmembrane receptors and their downstream signaling pathways [18,19,20,21]. GISTs are the most common mesenchymal tumors of the gastrointestinal tract and molecular diagnosis has transformed their management. In clinical practice, these diagnosis-defining mutations are detected in tumor DNA by Sanger sequencing or gene panel targeted next-generation sequencing.

#### 3.2.2. Inactivating Mutations

Several sarcomas are associated to inactivating mutations of tumor suppressor genes. As in most cancers, genes such as *TP53* and *PTEN* are frequently mutated during the course of oncogenesis [4,22,23], but some inactivating mutations constitute the primary molecular alteration. For instance, malignant peripheral nerve sheath tumors (MPNST) are characterized by mutations in the *NF1* tumor suppressor gene (50%) [24]. Perivascular epithelioid cell tumors (PEComas) are associated with mutations in *TSC1* and *TSC2* with subsequent activation of the mTOR pathway [25,26]. Another group of sarcomas, BAF-deficient sarcomas, harbor mutations in genes of the BAF (also called SWI-SNF) complex: epithelioid sarcomas [27] and malignant rhabdoid tumors including atypical teratoid/rhabdoid tumors (ATRTs) of the central nervous system (*SMARCB1* mutations) [28], small cell carcinomas of the ovary, hypercalcemic type (SCCOHT), and SMARCA4-deficient thoracic sarcomas (*SMARCA4* mutations) [29,30]. It has been shown recently that a subgroup of ATRTs have mutations of *SMARCA4*, and they are distinct from classical *SMARCB1*-mutated ATRTs [31]. The BAF complex is involved in chromatin remodeling and highlights the essential role of epigenetics in the pathogenesis of sarcomas. In clinical practice, these mutations can be found in tumor DNA by Sanger sequencing or gene panel targeted next-generation sequencing. Moreover, loss of proteins of the BAF complex can be shown using immunohistochemistry.

### 3.3. Gene Amplifications

A significant proportion of sarcomas harbor gene amplifications, the most frequent of which is the 12q amplification characteristic of adipocytic tumors: atypical lipomatous tumors (ALT) and well-differentiated liposarcomas (WDLPS) and dedifferentiated liposarcomas (DDLPS) [32]. Less often, the same amplification can be found in other tumors such as intimal sarcomas [33]. The 12q amplicon can be different in length and composition from one tumor to another, but it invariably contains the *MDM2* gene, which is an antagonist of *TP53*, and it promotes oncogenesis through suppression of the activity of the p53 protein [34], as well as through its direct binding to the chromatin to promote serine metabolism dependency [35]. DDLPS are tumors that contain two compartments: one is composed of adipocytic tumor cells and is similar to WDLPS, while the dedifferentiated compartment consists of undifferentiated high-grade tumor cells that may be confused with other high-grade non-lipogenic sarcomas such as undifferentiated pleomorphic sarcoma (UPS) or MPNST, or sometimes show heterologous differentiation with features of osteogenic or myogenic differentiation. Thus, *MDM2* amplification is an essential diagnostic tool to diagnose liposarcomas and in practice it can be found with FISH [36]. Other techniques that can be used are comparative genomic hybridization (CGH) and whole exome sequencing. When using these techniques, it is common to find a large number of genomic rearrangements in DDLPS [37], highlighting the limits of classifying sarcomas into sarcomas with simple or complex genetics.

## 4. Sarcomas with “Complex Genetics”

Genomically complex sarcomas represent more than 50% of soft tissue sarcomas in adults. In contrast to sarcomas with simple genetics, they do not harbor specific and characteristic molecular alterations. Indeed, they show large numbers of genomic rearrangements, copy number variations and point mutations, sometimes dubbed “genomic chaos”. While some recurrent mutations can be found in tumor suppressor genes such as *TP53*, *RB1*, and *ATRX* [4], molecular biology techniques are less essential for the diagnosis of these sarcomas, which are still predominantly defined by pathology associated to immunohistochemistry. However, it can still be of help in difficult situations, for instance in differentiating a benign from a similar-looking malignant tumor. One example is the distinction to be made between benign leiomyomas and malignant leiomyosarcomas in smooth muscle tumors of the uterus. Microscopic features such as mitoses and tumor necrosis are classically used to distinguish between benign and malignant tumors, but they may sometimes be difficult to assess, leading to the diagnosis of uterine smooth muscle tumors of unknown malignant potential (STUMPs). Genomic analysis with CGH array or whole exome sequencing can be used in these cases to detect malignant tumors that show a genomic index (score of genomic rearrangement) of more than ten [38].

## 5. Molecular Biology for Sarcoma Treatment

Molecular biology has allowed a refined classification of sarcomas with the definition of novel molecular entities and the identification of driver alterations in recent years. Independently of this diagnostic interest, the identification of characteristic molecular alterations in many sarcomas has allowed the improvement of clinical management, by a combination of better prognostication and therapeutic targeting based on knowledge of the underlying molecular changes in sarcomas (Table 2). This can also inform the design of clinical trials so that homogeneous groups of patients with similar prognoses and response to treatment are studied in trials, as opposed to a current pathological selection criteria that tends to group together different molecular entities, as well as the search for relevant biomarkers of response to treatment.

### 5.1. Gene Fusions

As detailed previously, Ewing sarcoma is a paradigmatic example of fusion-driven sarcoma with gene fusions between the FET and the ETS family genes [64]. Until recently, some sarcomas were classified as “Ewing-like” because of similar clinical and pathological characteristics, but presenting without FET–ETS fusion genes. RNA-seq has since allowed identification of several other fusion genes that have enabled the accurate classification of these Ewing-like tumors into several entities with similar characteristics in terms of prognosis, thus informing clinical management [65,66]. For instance, BCOR-rearranged sarcomas [67] are characterized by overall favorable prognosis, and they are treated with classical multimodal treatment [68], while *CIC*-rearranged sarcomas [69] have a much poorer prognosis, and they exhibit an intrinsic resistance to classical chemotherapy and radiotherapy regimes [70]. Independently of routine practice, this refinement of molecular classification has also allowed the homogenization of patients enrolled in clinical trials.

In children, one of the most frequent sarcoma subtypes is rhabdomyosarcoma, which was classically divided into alveolar and embryonal subtypes based on histopathological characteristics. With molecular biology, the former was found to be characterized by the presence of the oncogenic fusion *PAX3/PAX7-FOXO1*, which was a better indication than pathology for worse prognosis than sarcomas without the fusion [71]. With recent techniques of next-generation sequencing, rhabdomyosarcomas were further divided into sarcomas with *VGLL2* fusions [72] or *SRF* fusions [16] and overall favorable prognosis, *TFCP2* fusions and intermediary prognosis [12], and *MYOD1* mutations and worse prognosis [73]. This is an example of the clinical interest of identifying distinct molecular subgroups of differing prognoses as this may lead to altered clinical management and distinct inclusion into clinical trials.

Another example where molecular division can lead to different clinical management of previously similarly classified sarcomas are in endometrial stromal sarcoma (ESS). It has been shown that low-grade ESS is more often associated to *JAZF1* fusions [74], while high-grade ESS is characterized by fusions of genes from the 14-3-3 family such as *YWHAE* [75]. Recently, the use of RNA-seq showed that ESS could be divided into three groups based on fusion genes with different prognoses, once again showing the interest of using molecular biology for clinical management and design of clinical trials with homogeneous entities [76].

While gene fusions in sarcomas often involve transcription factors which are not easily druggable, some fusions involve tyrosine kinase receptors which are molecular drivers of oncogenesis that can be directly targeted for molecularly-driven therapy. The most striking example is the case of *NTRK*-fused sarcomas [14,77], for which treatment by NTRK inhibitors such as larotrectinib showed efficacy with a 79% overall response rate, median overall survival of more than 3 years in *NTRK*-fused tumors comprising about half of sarcomas (infantile fibrosarcoma, which is associated with *NTRK* fusions in over 90% of cases, but also osteosarcoma, chondrosarcoma, GIST), with a remarkably low toxicity [39,40]. For patients with sarcomas, the response rates were 74% and 94% for adult and pediatric patients, respectively, with progression-free and overall survival not reached at more than one year of follow-up. Larotrectinib has been approved in North America and Europe for *NTRK*-fused advanced and metastatic solid cancers, while a more recent NTRK inhibitor, entrectinib, has also been approved by the FDA [41].

Inflammatory myofibroblastic tumors (IMT) are mesenchymal tumors of intermediary malignant potential characterized by fusions involving the *ALK* gene in more than 50% of cases inducing the upregulation of this tyrosine kinase receptor. Other fusions involving the *ROS1* gene have been described in cases without *ALK* fusion [78]. Crizotinib, an ALK and ROS1 inhibitor, has shown a high response rate of 86% with a median response duration of more than 2.5 years in 14 patients with IMT [42]. This was further confirmed in a phase II trial with IMT patients of more than 15 years old, regardless of the presence of an *ALK* fusion, which showed response rates of 50% and 14% in patients harboring an *ALK* fusion or not, respectively [43]. Newer-generation ALK and ROS1 inhibitors including ceritinib (NCT02465528) and repotrectinib (NCT04094610) are currently evaluated in this indication.

Dermatofibrosarcoma protuberans (DFSP) is a cutaneous tumor of mesenchymal origin, and its intermediate malignant potential is characterized by recurrent fusion involving the genes *COL1A1* and *PDGFB*, leading to uncontrolled synthesis of a constitutively active form of the PDGFB growth factor. Imatinib, a tyrosine kinase inhibitor against BCR-ABL, KIT but also PDGF receptors, has shown response rates near 67% in several studies of DFSP patients with many complete responses [44,45]. Additionally, imatinib has been used as neoadjuvant therapy with a response rate between 40% and 60%, with a reduction in the tumor burden of approximately 20% [46,47].

Pigmented villonodular synovitis (PVNS), also known as diffuse-type tenosynovial giant cell tumor (TGCT), is a mesenchymal tumor affecting the joints and juxtaarticular soft tissues, characterized by infiltration by inflammatory cells; more than 50% of them are characterized by a gene fusion between *COL6A3* and *CSF1* [79], leading to overproduction of the CSF1 protein, which attract inflammatory cells expressing CSF1R [80]. Emactuzumab is a monoclonal antibody directed against CSF1R, and it was shown in a phase I study to have a response rate of 86% in 24 patients, with 2 complete responses [48]. Another antibody, cabiralizumab, also showed efficacy in a phase I/II study with some partial responses [81]. Pexidartinib is a tyrosine kinase inhibitor that targets CSF1R: in a phase III trial with 120 patients, the response rate was 39% versus 0% in the placebo group, with significant clinical benefit in patients [49]. Pexidartinib has thus been approved as a reference therapy for inoperable PVNS/TGCT.

While transcription factors are notoriously more difficult to target therapeutically, all oncogenic fusions in sarcomas are attractive targets for drug development, since they are the driver event of these diseases, and they have been successfully targeted as outlined above. More effective therapies targeting them are thus expected for the future [82,83].

### 5.2. Mutations

While sarcomas with mutations are less frequent than fusion-driven sarcomas, they are natural candidates for classical targeted precision medicine, with the paradigmatic successful example being GISTs. As outlined above, these tumors are characterized by mutations in the genes *KIT* (85%) and *PDGFRA* (15%). As early as 2002, imatinib showed efficacy in locally advanced and metastatic GISTs [52]. Since then, continued improvements have been made in the development of new inhibitors and understanding of drug resistance in GISTs. In approximately twenty years, overall survival in this disease has been transformed from approximately 8 months to more than 10 years for a large proportion of patients. The latest generation of drugs include avapritinib, which specifically targets mutations of resistance to imatinib, such as D842V [84]. In these imatinib-resistant patients, a response rate of 88% has been shown [85], leading to its accelerated FDA approval in this indication [86]. Avapritinib has shown higher response rates than regorafenib (17.1% vs. 7.2%) in a recent phase III trial including KIT-mutated patients after third-line treatment, however, this trial failed to reach its primary endpoint as the median progression-free survival was not significantly changed between both arms [50]. Ripretinib is another inhibitor of KIT and PDGFRA that showed in a randomized phase III trial a response rate of 9.4% and progression-free survival of 6.3 months, versus 1 month in placebo, after treatment by imatinib, sunitinib, and regorafenib [51]. It has thus become the standard fourth-line therapy for GISTs. Though it failed to show superiority over sunitinib in second-line after treatment by imatinib in a recent phase III trial yet to be presented, it was associated with less adverse events than sunitinib [55].

PEComas are characterized by *TSC1* and/or *TSC2* loss-of-function mutations leading to uncontrolled activation of the mTOR pathway [87]. Sirolimus, an mTOR inhibitor, induced partial responses in some patients with PEComa [88,89], achieving a response rate of 73% with 5-year overall survival of 65% in a series of 15 patients [58]. A multicenter trial with 31 patients treated by nab-sirolimus showed a response rate of 39% including 2 complete responses and 67% of responses lasting more than 12 months with a median overall survival of 40.8 months [56], leading to the approval of nab-sirolimus by the FDA for unresectable or metastatic PEComas.

SMARCB1-deficient sarcomas are mainly malignant rhabdoid tumors, epithelioid sarcomas but also some rare tumors such as chordomas. Inactivating mutations in *SMARCB1* lead to unregulated activity of the PRC2 histone methyltransferase complex and its EZH2 subunit. Tazemetostat is an oral selective EZH2 inhibitor that has been evaluated in pediatric SMARCB1-deficient sarcomas and adult epithelioid sarcomas. In the pediatric phase I trial, tazemetostat showed a response rate of 17% and median durations of response of more than 6 months in various histologies [59]. In a phase II trial of 62 adult patients with advanced epithelioid sarcoma, the response rate was 15% with median duration of response not reached after more than 1 year of follow-up; progression-free survival and overall survival were 5.5 and 19 months, respectively [60]. Based on these results, the FDA granted accelerated approval to tazemetostat for the treatment of advanced epithelioid sarcomas.

Some sarcomas harbor mutations in *IDH1* or *IDH2*, rendering them sensitive to PARP inhibition due to defective homologous recombination repair. In a recent study of IDH1/2-mutant tumors, three out of five chondrosarcomas and one epithelioid hemangioendothelioma derived clinical benefit from olaparib monotherapy [63].

### 5.3. Gene Amplifications

The most frequent sarcoma with gene amplification is liposarcoma which harbors *MDM2* amplification on the 12q amplicon. Numerous MDM2 inhibitors have been developed, and they have shown good activity in vitro and in xenografts [90,91]. However, phase I trials in humans showed response rates under 10% [61,62]. As the 12q amplicon also contains the *CDK4* gene, inhibitors such as palbociclib and ribociclib were tested but without significant evidence of activity [92,93]. A potential practice-changing phase III trial (NCT05218499) has just begun with BI-907828, an MDM2 inhibitor, against doxorubicin in first-line treatment [94]. Combinations of MDM2 and CDK4 inhibitors showed preclinical activity in liposarcomas [95], and they are also being tested in patients, though one recent study showed only three partial responses in 74 patients [96].

## 6. Perspectives

Sarcoma is a rapidly changing field for molecular biology, with novel techniques constantly discovering molecular underpinnings that may be targetable. While refined diagnosis is key, prognosis can also benefit from molecular biology. An example is the CINSARC signature which is a transcriptomic signature of 67 genes that reflects genomic complexity of tumors [97]. For operated sarcomas with complex genomics, it does better at predicting relapse than histological grade. It was also shown to be useful in other cancers such as GISTs, but also lymphomas and some carcinomas [98]. Neoadjuvant chemotherapy given based on the CINSARC signature is currently being evaluated in two clinical trials for resectable soft tissue sarcomas (NCT03805022, NCT0430727).

Though immunotherapy is still to make real progress in sarcomas [99], the importance of the immune microenvironment and its characterization by molecular techniques has been shown in a recent study using transcriptomic data: using a bioinformatics tool called MCP-counter [100] to estimate the abundance of several microenvironment cell populations in a cohort of more than 600 sarcomas including leiomyosarcomas, liposarcomas and undifferentiated pleomorphic sarcomas, the authors showed that sarcomas could be classified in five subgroups according to immune microenvironment, one of them showing enhanced immune infiltration by T and B lymphocytes as well as NK cells, and the presence of tertiary lymphoid structures (TLS) had better survival [101]. This has led for instance to the design of a phase II trial evaluating immune checkpoint combined inhibition for tumors with TLS (NCT04095208). As more studies are done to evaluate immunotherapy in sarcomas, it is expected that molecular determinants of response and biomarkers will be of paramount importance to predict response to immunotherapy, including single-cell techniques that can guide precision immunotherapy.

As for molecularly targeted therapy, trials such as the MULTISARC randomized multicenter study (NCT03784014) are evaluating the value in advanced soft tissue sarcomas of using next-generation sequencing (whole exome sequencing and RNA sequencing) to propose molecularly guided therapy based on targetable genomic alterations after first-line treatment [102].

As for diagnosis, it has been shown above that RNA sequencing is an interesting technique especially for fusion-driven sarcomas, and it is being increasingly used for diagnostic purposes. Another molecular technique is the use of DNA methylation arrays for the profiling of genome-wide DNA methylation changes. Combined with machine learning algorithms, this technique originally showed its value in refining classification and helping the diagnosis of central nervous system tumors [103]. More recently, the same team showed that this tool could also accurately classify tumors into 62 methylation classes of a broad range of adult and pediatric sarcomas [104]. The end-product of DNA and RNA is protein, and proteomics is also one technique that has promising potential for characterizing sarcomas [105].

While these state-of-the-art molecular assays do enable refined diagnosis and improved scientific knowledge, they do not necessarily lead to treatment recommendations, they are costly, and they are not necessarily available in all centers. It is likely that these techniques will continue to be used only in dedicated reference centers, until diminishing costs, improved techniques, or stronger clinical impact can be achieved. For instance, RNA-seq is for now restricted to specialized centers, but it can be expected that as sequencing costs decrease and more centers have the means to perform RNA-seq, including on formalin-fixed, paraffin-embedded samples, this technique can be widely adopted and provide more data to demonstrate clinical practice-changing potential.

Finally, very recent technological breakthroughs have seen the advent of single-cell technologies, allowing the characterization of molecular features at the single-cell level and thus vastly increasing the capability to understand pathogenesis, tumor heterogeneity, interactions with the microenvironment, and resistance to treatment. Though the use of these costly single-cell technologies is still in its infancy for rare tumors such as sarcoma, they has been used to better characterize some tumors such as synovial sarcoma, Ewing sarcoma, and osteosarcoma [106,107,108]. It could be possible one day to use single-cell RNA-seq in routine to characterize tumor heterogeneity as prognostic or predictive biomarkers, as well as resistance mechanisms to therapy, especially immunotherapy [109,110].

## 7. Conclusions

Molecular biology has a prominent role in the diagnostic and therapeutic management of numerous soft tissue sarcoma subtypes. New techniques and future research hold great promise to tackle tumor heterogeneity and treatment resistance through individualized precision medicine strategies. 

## Figures and Tables

**Table 1 cancers-14-02548-t001:** Examples of chromosomal translocations and associated gene fusions most frequently detected in soft tissue sarcomas and mesenchymal tumors.

Sarcoma Subtype	Translocation	Genes	Oncogenic Mechanism
Ewing sarcoma	t (11; 22) (q24; q12)	*EWSR1, FLI1*	Transcription factor
	t (21; 22) (q22; q12)	*EWSR1, ERG*	
	t (16; 21) (p11; q22)	*FUS, ERG*	
DSRCT	t (11; 22) (p13; q12)	*EWSR1, WT1*	Transcription factor
Alveolar rhabdomyosarcoma	t (2;13) (q35; q14)	*PAX3, FOXO1*	Transcription factor
	t (1; 13) (p36; q14)	*PAX7, FOXO1*	
Clear cell sarcoma	t (12; 22) (q13; q12)	*EWSR1, ATF1*	Transcription factor
Extraskeletal myxoid chondrosarcoma	t (9; 22) (q22–31; q11–12)	*EWSR1, NR4A3*	Transcription factor
Myxoid liposarcoma	t (12; 22) (q13; q12)	*EWSR1, CHOP*	Transcription factor
	t (12; 16) (q13; p11)	*FUS, CHOP*	
Alveolar soft part sarcoma	t (X; 17) (p11.2; q25)	*ASPL, TFE3*	Transcription factor
PEComa	Xp11 rearrangement	**, TFE3*	Transcription factor
Low grade fibromyxoid sarcoma	t (7; 16) (q33; p11)	*FUS, CREB3L2*	Transcription factor
Sclerosing epithelioid fibrosarcoma	t (11; 22) (p11; q12)	*EWSR1, CREB3L1*	Transcription factor
Low grade endometrial stromal tumor	t (7; 17) (p15; q21)	*JAZF1, JJAZ1*	Transcription factor
Synovial sarcoma	t (X; 18) (p11; q11)	*SYT, SSX1, SSX2, SSX4*	Chromatin remodeling
Congenital fibrosarcoma	t (12; 15) (p13; q25)	*ETV6, NTRK3*	Tyrosine Kinase
Inflammatory myofibroblastic tumor	t (2; 19) (p23; p13.1)	*TPM4, ALK*	Tyrosine Kinase
	t (1; 2) (q22–23; p23)	*TPM3, ALK*	
Dermatofibrosarcoma protuberans	t (17; 22) (q22; q13)	*COL1A1, PDGFβ*	Growth Factor
PVNS/TGCT	t (1; 2) (p13; q37)	*COL6A3, CSF1*	Growth Factor

Abbreviations: DSRCT: desmoplastic small round cell tumor, PVNS/TGCT: pigmented villonodular synovitis/tenosynovial giant cell tumor. * Multiple gene partners.

**Table 2 cancers-14-02548-t002:** Examples of molecularly targeted therapies in soft tissue sarcomas.

Sarcoma Subtype	Molecular Target	Targeted Therapy	Overall Response Rate	Clinical Use	References
NTRK-fused sarcoma	*NTRK1/2/3* fusions	Entrectinib	46%	FDA-approved	[39,40,41]
			74% (adults)		
		Larotrectinib	94% (children)	FDA-approved	
IMT	*ALK/ROS* fusions	Crizotinib	50% (adults)	Off-label use	[42,43]
			85% (children)		
DFSP	*COL1A1-PDGFB* fusion	Imatinib	67% (advanced phase)	FDA-approved	[44,45,46,47]
			40–60% (neoadjuvant)		
PVNS/TGCT	*COL6A3-CSF1* fusions	Emactuzumab	86%	Not approved	[48,49]
		Pexidartinib	39%	FDA-approved	
GIST	*KIT* mutations	Imatinib	53.7%	FDA-approved	[50,51,52,53,54]
		Sunitinib	7%	FDA-approved	
		Regorafenib	4.5%	FDA-approved	
		Ripretinib	9.4% (4th line)	FDA-approved	
		Avapritinib	17.1% (>2nd line)	Not approved	
GIST	PDGFRA D842V mutation	Avapritinib	88%	FDA-approved	[55]
PEComa	*TSC1/2* mutation	Everolimus	41%	Off-label use	[56,57,58]
		Sirolimus	73%	Off-label use	
		Nab-sirolimus	39%	FDA-approved	
Epithelioid sarcoma	*SMARCB1* mutation	Tazemetostat	15% (adults)	FDA-approved	[59,60]
MRT/ATRT			17% (children)		
DDLPS	*MDM2* amplification	SAR405838	<10%	Not approved	[61,62]
		MK-8242			
Mesenchymal sarcomas	*IDH1-2* mutation	Olaparib	17%	Not approved	[63]

Abbreviations: IMT: inflammatory myofibroblastic tumors; DFSP: dermatofibrosarcoma protuberans; PVNS/TGCT: pigmented villonodular synovitis/tenosynovial giant cell tumor; GIST: gastrointestinal stromal tumors, MRT: malignant rhabdoid tumors; ATRT: atypical teratoid rhabdoid tumors; DDLPS: dedifferentiated liposarcoma.

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
