# Peer review of "The Molecular Biology of Soft Tissue Sarcomas: Current Knowledge and Future Perspectives"

_cancers, 2022, doi:10.3390/cancers14102548_

Round 1
Reviewer 1 Report
The authors Vibert and Watson present a good written overview review on molecular diagnostic, alterations and implications on treatments in soft tissue sarcoma (STS). They first give a general overview about genetic alterations and the typical separation in STS with defined “simple” genetic alterations and in STS with “complex genetics”. Then they describe common translocation, mutations and amplifications in STS. They also have a small paragraph on “complex genetics” with main focus on uterine leiomyoma, STUMP and LMS. In the second part they discuss the impact of molecular alteration on treatment and targeted therapy in STS. The review ends with a paragraph on perspectives.
The review gives a good basic overview on the topic for a broader readership and addresses an interesting and emerging topic. However, major concerns remain before recommendation for publication in cancers:
- I miss some critical discussion in the review. Especially the balancing act between -to say it provocative- perform RNA Seq for all STS with extensive costs and the still limited impact on treatment in many cases. From research perspective, it is very interesting, but might also not available at many places.
- The paragraph on perspectives appears a bit biased since 1/3 is about 3 studies at which the institute is involved, where the authors work, too.
- The authors should state more clearly, especially in table 2, which drugs are approved, which are common off-label use, which are still under clinical investigation. E.g. pexidartinib is only approved by FDA, but not EMA. Crizotinib is not approved.
- Regarding the statements on Avapritinib corrections are required: Avapritinib is not approved for KIT-mutated GIST. Moreover, in the phase III trial on Avapritinib vs. Regorafenib, the primary endpoint was not met with no difference in PFS. This is misleading in the review and should be stated more clearly.
- Why do the authors not mention in table 2 Imatinib, Sunitinib and Regorafenib for KIT-mutated GIST?
- On p. 8 l. 307 “could” should be replaced by “is”, since Ripretinib is approved 4th line therapy in KIT-mutated GIST.
- Why do the authors not mention classical sirolimus and everolimus for PEComa in table 2?
- I thinks it is worth to mention more clearly on p. 8 that tazemetostat is approved for epitheloid sarcoma by FDA.
- Findings from Eder et al. on clinical efficacy of Olaparib in IDH1/IDH2-mutant mesenchymal sarcomas should be included and discussed (DOI https://doi.org/10.1200/PO.20.00247).
Author Response
The authors Vibert and Watson present a good written overview review on molecular diagnostic, alterations and implications on treatments in soft tissue sarcoma (STS). They first give a general overview about genetic alterations and the typical separation in STS with defined “simple” genetic alterations and in STS with “complex genetics”. Then they describe common translocation, mutations and amplifications in STS. They also have a small paragraph on “complex genetics” with main focus on uterine leiomyoma, STUMP and LMS. In the second part they discuss the impact of molecular alteration on treatment and targeted therapy in STS. The review ends with a paragraph on perspectives.
The review gives a good basic overview on the topic for a broader readership and addresses an interesting and emerging topic. However, major concerns remain before recommendation for publication in cancers:
- I miss some critical discussion in the review. Especially the balancing act between -to say it provocative- perform RNA Seq for all STS with extensive costs and the still limited impact on treatment in many cases. From research perspective, it is very interesting, but might also not available at many places.
We thank the reviewer for this comment, which indeed raises a critical point for the use of molecular assays in the clinical management of sarcomas. We have thus added the following paragraph in the last part of the article (lines 419-427):
While these state-of-the-art molecular assays do enable refined diagnosis and improved scientific knowledge, they do not necessarily lead to treatment recommendations, are costly and not necessarily available in all centers. It is likely that these techniques will continue to be used only in dedicated reference centers, until diminishing costs, improved techniques or stronger clinical impact can be achieved. For instance, RNA-seq is for now restricted to specialized centers, but it can be expected that as sequencing costs decrease and more centers have the means to perform RNA-seq, including on formalin-fixed, paraffin-embedded samples, this technique can be widely adopted and provide more data to demonstrate clinical practice-changing potential.
- The paragraph on perspectives appears a bit biased since 1/3 is about 3 studies at which the institute is involved, where the authors work, too.
We agree that the three studies mentioned in this paragraph (ie CIRSARC, PEMBROSARC and MULTISARC) are being conducted in France within the French Sarcoma Group and that our institution is involved in these studies (as investigators, however not as PI). However, to our knowledge, these are the only randomized clinical trials dedicated to investigating immune signature and impact of next generation sequencing in soft tissue sarcomas. Thus, we believe that it is necessary to mention them here.
- The authors should state more clearly, especially in table 2, which drugs are approved, which are common off-label use, which are still under clinical investigation. E.g. pexidartinib is only approved by FDA, but not EMA. Crizotinib is not approved.
A dedicated column for the current clinical use has been added into Table 2 with the status of the drugs
- Regarding the statements on Avapritinib corrections are required: Avapritinib is not approved for KIT-mutated GIST. Moreover, in the phase III trial on Avapritinib vs. Regorafenib, the primary endpoint was not met with no difference in PFS. This is misleading in the review and should be stated more clearly.
The sentence has been rephrased to make it less misleading (lines 335-338)
Avapritinib has shown higher response rates than regorafenib (17.1 % vs 7.2 %) in a recent phase III trial including KIT-mutated patients after third-line treatment, however this trial failed to reach its primary endpoint as the median progression-free survival was not significantly changed between both arms [50]
- Why do the authors not mention in table 2 Imatinib, Sunitinib and Regorafenib for KIT-mutated GIST?
Imatinib, sunitinib, and regorafenib have been added to Table 2.
- On p. 8 l. 307 “could” should be replaced by “is”, since Ripretinib is approved 4th line therapy in KIT-mutated GIST.
This has been modified accordingly
- Why do the authors not mention classical sirolimus and everolimus for PEComa in table 2?
Everolimus and sirolimus have been added to Table 2
- I thinks it is worth to mention more clearly on p. 8 that tazemetostat is approved for epitheloid sarcoma by FDA.
The following sentence has been added (lines 363-4):
Based on these results, FDA granted accelerated approval to tazemetostat for the treatment of advanced epithelioid sarcomas.
- Findings from Eder et al. on clinical efficacy of Olaparib in IDH1/IDH2-mutant mesenchymal sarcomas should be included and discussed (DOI https://doi.org/10.1200/PO.20.00247).
We thank the reviewer for this suggestion, we have added the following sentence and cited this study in the text (lines 365-8) and Table 2:
Some sarcomas harbor mutations in IDH1 or IDH2, rendering them sensitive to PARP inhibition due to defective homologous recombination repair. In a recent study of IDH1/2-mutant tumors, three out of five chondrosarcomas and one epithelioid hemangioendothelioma derived clinical benefit from olaparib monotherapy [63].
Reviewer 2 Report
Congratulation for the accurate and comprehensive review on molecular biology of soft tissue sarcoma. Pathologic diagnosis in sarcoma is an integration of morphologic , immunohistochemical and molecular characteristics.
I have only few comments/questions:
1- for me it's not clear the concept of "driver alteration". In paragraph "2. Molecular biology for sarcoma diagnosis", line 64 "" Which are the groups of sarcoma with a driver alteration? Are all the sarcomas with "simple genetics"? And, are all the gene fusions (listed in table 1) drivers of sarcomagenesis?
2-the paragraph "3.3. Activating mutations" and "3.4. Inactivating mutations " are sub-paragraphs of "3.2. Mutations"? If yes, they should be respectively numbered as 3.2.1 and 3.2.2 . And "3.5. Gene amplifications " will be 3.3
Author Response
Congratulation for the accurate and comprehensive review on molecular biology of soft tissue sarcoma. Pathologic diagnosis in sarcoma is an integration of morphologic , immunohistochemical and molecular characteristics.
I have only few comments/questions:
1- for me it's not clear the concept of "driver alteration". In paragraph "2. Molecular biology for sarcoma diagnosis", line 64 "" Which are the groups of sarcoma with a driver alteration? Are all the sarcomas with "simple genetics"? And, are all the gene fusions (listed in table 1) drivers of sarcomagenesis?
We thank reviewer 2 for this comment. Indeed the concept of “driver alteration” remains debatable, since the oncogenic properties of all the fusion genes found in soft tissue sarcoma have not yet been assessed, especially in rare subtypes. However, their similarities in terms of structure (ie chimeric transcription factors or chimeric tyrosine kinase receptors), the homogeneity of the gene expression profiles of tumors harboring a given fusion gene, as well as the scarcity of other molecular alterations (such as mutations and copy number alterations) in these tumors are consistent with the fact that these gene fusions are a very early (and probably the first) event in the oncogenesis of these tumors.
This comment has been addressed in the manuscript (lines 59-64)
If the oncogenic properties of all the gene fusions found in rare sarcomas have not yet been assessed in relevant models, their similarities in terms of structure, the homogeneity of the gene expression profiles of tumors with a given fusion, as well as the scarcity of other genomic alterations found in their genomes, suggest that these molecular alterations are a very early and driver event in the oncogenesis of these tumors.
2-the paragraph "3.3. Activating mutations" and "3.4. Inactivating mutations " are sub-paragraphs of "3.2. Mutations"? If yes, they should be respectively numbered as 3.2.1 and 3.2.2 . And "3.5. Gene amplifications " will be 3.3
This has been modified accordingly.
Reviewer 3 Report
The authors outline a well-written and concise review about the current understanding of the molecular biology of sarcomas.
Minor things should be discussed/edited for publication:
paragraph 3:
- It would be nice if the authors would change the UPS part to non-lipogenic in their discussion of DDLPS. That component often shows heterologous differentiation (such as osteosarcoma, myogenic etc) which should be included here.
- the differentiation between leiomyoma and leiomyosarcoma can be tough and CGH may serve as an adjunct but other (histopathologic) features are usually use to make this distinction (mitoses, tumor necrosis). This should be made clear.
- p3 table: add “extraskeletal” in front of myxoid chondrosarcoma; important distinction!
- add “diffuse type” behind tenosynovial giant cell tumor in line 276
Wording/typos:
- line 226: take out the word “out”
- line 158: “can” instead of “car”
- please italicize all gene names - does not appear consistent
Author Response
The authors outline a well-written and concise review about the current understanding of the molecular biology of sarcomas.
Minor things should be discussed/edited for publication:
paragraph 3:
- It would be nice if the authors would change the UPS part to non-lipogenic in their discussion of DDLPS. That component often shows heterologous differentiation (such as osteosarcoma, myogenic etc) which should be included here.
The sentence has been rephrased to include this comment (lines 179-183):
« while the dedifferentiated compartment consists of undifferentiated high-grade tumor cells that may be confused with other high-grade non-lipogenic sarcomas such as undifferentiated pleomorphic sarcoma (UPS) or MPNST, or sometimes show heterologous differentiation with features of osteogenic or myogenic differentiation. »
the differentiation between leiomyoma and leiomyosarcoma can be tough and CGH may serve as an adjunct but other (histopathologic) features are usually use to make this distinction (mitoses, tumor necrosis). This should be made clear.
The sentence has been modified accordingly to include this comment (lines 198-203):
One example is the distinction to be made between benign leiomyomas and malignant leiomyosarcomas in smooth muscle tumors of the uterus. Microscopic features such as mitoses and tumor necrosis are classically used to distinguish between benign and malignant tumors but may sometimes be difficult to assess, leading to the diagnosis of uterine smooth muscle tumors of unknown malignant potential (STUMPs).
p3 table: add “extraskeletal” in front of myxoid chondrosarcoma; important distinction!
This has been modified accordingly.
add “diffuse type” behind tenosynovial giant cell tumor in line 276
This has been modified accordingly.
Wording/typos:
- line 226: take out the word “out”
- line 158: “can” instead of “car”
- please italicize all gene names - does not appear consistent
All typos have been corrected and gene names have been italicized.
Round 2
Reviewer 1 Report
The concerns are well adressed and I see major improvment of the manuscript. Therefore, a can recommend publication in cancers.